# Decreased Innate Migration of Pro-Inflammatory M1 Macrophages through the Mesothelial Membrane Is Affected by Ceramide Kinase and Ceramide 1-P

**DOI:** 10.3390/ijms232415977

**Published:** 2022-12-15

**Authors:** Chee Wai Ku, Joan Yang, Hong Ying Tan, Jerry Kok Yen Chan, Yie Hou Lee

**Affiliations:** 1Department of Reproductive Medicine, KK Women’s and Children’s Hospital, Singapore 229899, Singapore; 2Obstetrics and Gynaecology Academic Clinical Program, Duke-NUS Medical School, Singapore 169857, Singapore; 3KK Research Centre, Singapore 229899, Singapore; 4Yong Loo Lin School of Medicine, National University of Singapore, Singapore 117597, Singapore

**Keywords:** endometriosis, macrophage, ceramide, ceramide-1-phosphate, peritoneal, migration

## Abstract

The retrograde flow of endometrial tissues deposited into the peritoneal cavity occurs in women during menstruation. Classically (M1) or alternatively (M2) activated macrophages partake in the removal of regurgitated menstrual tissue. The failure of macrophage egress from the peritoneal cavity through the mesothelium leads to chronic inflammation in endometriosis. To study the migration differences of macrophage phenotypes across mesothelial cells, an in vitro model of macrophage egress across a peritoneal mesothelial cell monolayer membrane was developed. M1 macrophages were more sessile, emigrating 2.9-fold less than M2 macrophages. The M1 macrophages displayed a pro-inflammatory cytokine signature, including IL-1α, IL-1β, TNF-α, TNF-β, and IL-12p70. Mass spectrometry sphingolipidomics revealed decreased levels of ceramide-1-phosphate (C1P), an inducer of migration in M1 macrophages, which correlated with its poor migration behavior. C1P is generated by ceramide kinase (CERK) from ceramide, and blocking C1P synthesis via the action of NVP231, a specific CERK chemical inhibitor, prohibited the emigration of M1 and M2 macrophages up to 6.7-fold. Incubation with exogenously added C1P rescued this effect. These results suggest that M1 macrophages are less mobile and have higher retention in the peritoneum due to lower C1P levels, which contributes to an altered peritoneal environment in endometriosis by generating a predominant pro-inflammatory cytokine environment.

## 1. Introduction

The peritoneal cavity is covered by the peritoneum, comprising a single layer of flat, simple, stratified squamous epithelial cells, the mesothelium, which sits on a basal laminar. The microvilli on the surface of mesothelial cells secrete a protective barrier, called the glycocalyx, which is composed of glycosaminoglycans (GAGs) and provides a non-adhesive and protective surface for intracoelomic organs and tissues [1,2]. Within the peritoneal cavity, there are peritoneal macrophages that play a key role in the control of infection and inflammatory pathologies [3], as well as in maintaining the robustness of the immune response [4]. Inflammation resolution involves macrophage migration across the peritoneum to lymphatic vessels [5,6]. This occurs in the mesothelium that overlies the draining lymphatics. In vivo, macrophages adhere to the mesothelium [5], and the removal of inflammatory macrophages involves migration across the mesothelium to the draining lymphatics rather than local apoptosis [6]. The balance of the accumulated macrophage subset determines the fate of the further development or resolution of inflammation [7]. The dysregulation of macrophage migration and imbalances in the types of macrophages results in chronic inflammatory conditions such as endometriosis [8,9,10].

Endometriosis is a common chronic estrogen-dependent disease that affects 6–12% of women, mainly during their reproductive years, and is characterized by the presence of ectopic endometrial implants primarily in the pelvic peritoneum, ovaries, and rectovaginal septum [11]. Retrograde menstruation is thought to be the chief initiation event of endometriosis, and subsequent pathophysiology involves aberrant implantation, inflammation, and angiogenesis, eventually manifesting as symptomatic pain and infertility [12]. These symptoms have been attributed to the chronic inflammatory state of the pelvic peritoneal area with an altered immunological and inflammatory environment in the micro- and macroenvironment [13]. The microenvironment refers to the immediate small-scale environment of a cell, group of cells, or tissue, especially as a distinct part of a larger environment, known as the macroenvironment. The most widely accepted explanation for endometriosis is the immune evasion and implantation of foreign endometrial cells after retrograde menstruation in the presence of factors within the peritoneal cavity that stimulate cell growth. Although retrograde menstruation is a common event, only a relatively small proportion of women develop endometriosis, suggesting that other factors are involved, for example, immunological and intraperitoneal biochemical alterations [14]. In a recent study in 2019 [15], it was revealed that ectopic endometrium could reduce the typical phagocytic ability of peritoneal macrophages in women with endometriosis through the modulation of SIRP-α and CD36 expression. As such, this causes inadequate removal of retrograde endometrial debris and an enhanced ability of endometrial lesions to implant on the peritoneal surface. In addition, the levels of macrophages and inflammatory cytokines are elevated in the peritoneal fluid of women with endometriosis, contributing to the chronically inflamed peritoneal environment [16]. Two well-appreciated macrophage polarization programs in vitro are classically activated (M1) and alternatively activated (M2) macrophages. M1 is heavily involved in the persistence of inflammation, while M2 is suspected to enable the establishment of the initial endometrial lesion. Macrophage activation enforces regenerative and vascular responses that aid tissue repair and restitution. Counterproductively, unregulated macrophage activation and response can be detrimental when the cause of the homeostasis disruption cannot be eliminated, such as in the case of endometriosis [17]. The altered and chronically inflamed peritoneal environment in endometriosis is associated with elevated levels of inflammatory cytokines and angiogenic factors, which are M1- and M2-associated, respectively [10]. Notably, the role of macrophages in inflammation has been implicated in other diseases such as peritonitis, inflammatory bowel disease, and abdominal cancers that grow or metastasize in the abdominal cavity [18]. Further studies, however, are required to elucidate the specific roles and pathophysiology of the different macrophage subtypes.

Sphingolipid metabolism is dysregulated in endometriosis. Various types of sphingolipids, including glucosylceramide (GlcCer) and ceramide (Cer), were found to be accumulated in the serum and PF of endometriosis patients [19]. Of these sphingolipids, Cer is a central intermediate compound in sphingolipid metabolism that can be metabolized to ceramide 1-phosphate (C1P) by the metabolic enzyme ceramide kinase (CERK). As such, women with endometriosis have increased C1P in their peritoneal fluid as compared to women without the disease [19]. C1P and CERK are known to promote the migration of hematopoietic stem progenitor cells, smooth muscle cells, multipotent stromal cells, human umbilical vein endothelial cells, coronary artery macrovascular endothelial cells, retinal microvascular endothelial cells, and mouse macrophage cell lines [20,21]. The presence of an inflammation response induces the recruitment of monocytes through the mesothelium, which in turn differentiates into macrophages and their subsequent egress upon inflammation resolution [22]. The migration of C1P occurs through pathways involving Akt and kappa B, stimulating the release of monocyte chemoattractant protein-1 (MCP-1) in vitro [20,23].

In this study, we aimed to ascertain the differences in the migratory properties of macrophage phenotypes across mesothelial cells in vitro and the correlation with C1P levels. Several in vitro models include 2D and more complex 3D cultures of endometrial stromal and peritoneal mesothelial cells [24,25,26]. However, there are few in vitro models that enable the interaction of macrophage and peritoneal mesothelial cells. Using an in vitro peritoneal mesothelium model in which macrophages were seeded on a monolayer of mesothelial cells, we observed that M2 macrophages emigrated more than M1 across the mesothelial monolayer, with higher C1P levels increasing the migration rate. In contrast, M1 macrophages were retained and secreted a variety of pro-inflammatory cytokines, shedding light on the role of ceramide-C1P metabolism in amplified and persistent peritoneal inflammation in endometriosis. Therefore, we hypothesized that C1P and its metabolic enzyme CERK influence the macrophage phenotypes to either remain in the peritoneal membrane in vitro and further contribute to inflammation or emigrate away. Hence, targeting the sphingolipid metabolism of macrophage phenotypes in endometriosis represents a potential novel pharmacological approach [27].

## 2. Results

### 2.1. M1 Macrophages Have Lower Innate Migration across the Mesothelium In Vitro

To evaluate macrophage retention in the peritoneum, we established an in vitro model of MeT-5A peritoneal mesothelial cells to mimic the peritoneal membrane (Figure 1A). At full confluency, the MeT-5A mesothelial cells formed uninterrupted monolayers overlying a collagen-coated transwell membrane, resembling the apical membranous mesothelial monolayer and its underlying basal lamina of the peritoneum (Figure 1B). The mesothelial monolayer was stained and appeared positive for mesothelin, a marker of mesothelial cells (Appendix A) [28].

Using the developed peritoneum in vitro migration model, the innate migratory rate of polarized macrophages was assessed and compared. Monocytes were polarized to the M0, M1, and M2 macrophages. To recapitulate the egress of macrophages out of the peritoneum and into the lymphatic system, polarized macrophages were added to the upper transwell chamber (analogous to the peritoneal membrane); allowed to emigrate to the lower chamber (analogous to the lymphatic space) over 20 h; and counted. After 20 h, a larger proportion of M1 macrophages were retained in the upper chamber compared to M2. When normalized to unactivated M0 macrophages, the M1:M0 ratio was significantly (2.9-fold) lower than the M2:M0 ratio (Student’s two-tailed *t*-test, *p* < 0.01; Figure 1C and Appendix A). Conversely, due to the weaker migratory property of M1 macrophages, more cells were retained on the membrane. To eliminate potential bias in migration due to M1 and M2 macrophage size differences [29], we repeated the experiment using a larger pore size of 8 µm and demonstrated significantly more M2 macrophage emigration (5.9-fold increase, *p* < 0.005; Appendix A). Additionally, macrophages incubated with C1P demonstrated enhanced migration (12.9-fold increase, *p* < 0.005; Figure 1D). Together, these results suggested a lower innate migration rate of M1 relative to M2 macrophages.

### 2.2. Retained M1-Polarized Macrophages Have a Pro-Inflammatory Cytokine Profile

To evaluate the state of the inflammatory milieu of retained macrophages, the secreted cytokines were evaluated using a multiplex immunoassay. Multiple cytokines interact to orchestrate immune and other complex responses, together with immunomodulatory proteins that form signalling networks associated with clinical outcomes [16,30]. Multivariate statistical analysis using partial least squares regression (PLSR) could identify significant associations between cytokines that act together in tandem with the macrophage subtype. In PLSR, the collection of cytokine measurements forms a matrix that is transformed into a principal component space, where principal components are a set of orthogonal coordinates that maximizes the covariation in the cytokine data matrix and the outcome dataset. According to multivariate PLSR, the cytokines delineated M0, M1, and M2 macrophages (Figure 2A), with 44% of the covariance explained by principal component 1 and 81% of the covariance explained by principal component 2, implying a distinct cytokine signature for each phenotype. The influence of their distinctive cytokine profiles was further supported by the PLSR loadings plot, which yielded a large separation between the three phenotypes (in red) (Appendix A). Univariate statistical analysis revealed that IL-1α, IL-1β, IL-12p70, IP-10, eotaxin, TNFα, and TNFβ were significantly increased in M1 macrophages compared to M2 macrophages (Figure 2B). In particular, these significant cytokines overlapped with those determined by multivariate PLSR β-coefficients (IL-1α, IL-1β, IL-2, IL-7, IL-15, IL-27, FGF2, TNFα, and TNFβ) (Appendix A). On the contrary, the established M2 cytokines IL-10 and VEGF-A, together with IL-18 and LIF, were found to be significantly elevated in M2 macrophages compared to M1 macrophages (Figure 2B; Appendix A). Taken together, our data suggested that the lower innate migration rate of M1 macrophages led to their retention and the production of a strongly pro-inflammatory cytokine milieu.

### 2.3. Ceramide-1-Phosphate Regulates Macrophage Migration

C1P is a migration-modulatory lipid [19]. We hypothesized that under different polarization states, macrophage C1P levels are differentially produced, which in turn modulates the macrophages’ migratory behavior [19,22]. The immunoblot analysis showed the upregulated expression of the CERK protein that generates C1P from ceramide in M2 relative to M1 (Figure 3A), consistent with the upregulated expression of the CERK gene in M2-polarized macrophages [31]. Liquid chromatography–tandem mass spectrometry (LC-MS/MS) sphingolipidomics revealed the de novo synthesis of intracellular C1Ps, with increased C_16:0_, C_18:1_, and C_24:0_ C1P levels in M2 macrophages relative to M1 and M0 macrophages (Figure 3B). M2 macrophages also produced significantly higher C_16:0_, C_18:1_, C_24:0_, and C_24:1_ Cer levels than M1 macrophages, overlapping almost identically with the C1P increases (Figure 3C). The extracellular levels of C1P and Cer for M1 and M2 macrophages were similar, although M0 macrophages secreted greater amounts of C1P and Cer (Appendix A). The intracellular and extracellular S1P levels were also similar across the macrophage subtypes (Appendix A).

To further test our hypothesis that C1P stimulates cell migration, we blocked the synthesis of C1P through the action of NVP231, a specific small molecular inhibitor of CERK [32]. LC-MS/MS confirmed the pharmacological inhibition of C1P levels in macrophages by NVP-231 (M0 and M2: 2-fold increase relative to control; M1: 1.7-fold increase relative to control; Appendix A). The emigration of M1 macrophages was abolished in the presence of 500 and 1000 nM NVP-231; conversely, they were retained on the upper side of the membrane. Similarly, NVP231 reduced M2 macrophage migration in a dose-dependent manner, and at 1000 nM NVP-231, M2 macrophages emigrated at 6.7-fold lower rates across the transwell membrane compared to the control (*p* < 0.01; Figure 3D). Incubation with exogenously added C1P rescued this effect (*p* < 0.05; Figure 3C). Collectively, C1P modulated macrophage migration across the membrane, and the blockade of C1P metabolism attenuated macrophage egress (Figure 4).

## 3. Discussion

Endometriosis is a heterogeneous disease characterised by the presence of endometrial stromal glands outside of the uterus, including the peritoneum and ovaries. It is a debilitating disease with associated chronic pelvic pain and subfertility that negatively affect the biological, psychological, and social wellbeing of a patient [33]. The clinical presentation of endometriosis is largely identified as chronic inflammation in the pelvic cavity. However, the exact pathophysiology of endometriosis remains unknown. Previous studies have established the critical role of peritoneal macrophages in the pathophysiology of endometriosis. In fact, women with endometriosis have more peritoneal macrophages, and these macrophages have been found to be the most abundant immune cells present within endometriotic lesions [34]. In this study, we showed that the classically activated subtype (M1) has a pro-inflammatory cytokine profile and a lower innate migratory rate due to an impaired CERK/C1P metabolism, while the alternatively activated macrophage subtype (M2) has an anti-inflammatory profile, with a higher innate migratory rate.

In this study, M1 macrophages were associated with a pro-inflammatory cytokine profile, with the production of pro-inflammatory cytokines such as IL-1α, IL-1β, IL-12p70, IP-10, eotaxin, TNFα, and TNFβ. Similarly, we found that there was an increase in IL-1β mRNA levels in patients with endometriosis compared to those without endometriosis during the menstrual and luteal phases [35]. IL-1α and IL-1β cytokines are pro-inflammatory [36], with IL-1β being the most important mediator of acute and chronic inflammation and immune response. The aberrant control of IL-1β activity can enhance the risk of the adhesion and growth of the ectopic endometrium, leading to the development of endometriosis [37,38]. Furthermore, IL-1β may facilitate endometrial cell proliferation and endometrial-peritoneal angiogenesis, which in turn aids the development of endometriotic lesions [39]. In another study [40], it was revealed that a higher level of proIL-1β was present in endometriosis, and this abnormal proIL-1β concentration in the peritoneum increased inflammation and endometriosis formation. ProIL-1β also plays a role in the neovascularization that surrounds endometriotic lesions by increasing IL-6 and VEGF [41]. In addition, TNF-α and TNF-β were detected at significantly higher concentrations in M1 macrophages than in M2. TNF-α is a pro-inflammatory cytokine produced mainly by activated macrophages. It has been observed that the concentration of TNF-α is elevated in the peritoneal fluid of women with endometriosis and may correlate with the stage of the disease [42]. TNF-α also plays a critical role in local and systemic manifestations of endometriosis, as it stimulates the proliferation of endometriotic stromal cells by inducing IL-8 expression [43] and improves the adhesion of endometrial stromal cells to mesothelial cells [44].

M2 macrophages have displayed 2.9-fold higher migratory rates than M1 in in vitro cell migration assays [45,46], a finding that is supported by the results presented herein. A plausible explanation is the upregulation of CERK and C1P levels in M2 macrophages. At the protein level, the positive correlation between C1P levels and M2 polarization confers the increased migratory properties of M2. This is supported by other studies that have illustrated increased cell migration with added exogenous C1P [20]. C1P promotes cell migration by promoting the release of monocyte chemoattractant protein-1 (MCP-1). MCP-1, also known as chemokine ligand 2, induces the migration of monocytes through interactions with its receptor CCR2. This receptor couples with Gi proteins and causes the extracellularly regulated phosphorylation of kinases 1 and 2 and protein kinase B (Akt) upon ligation with C1P. Furthermore, C1P stimulates the DNA binding activity of the nuclear factor kappa B, which also plays a critical role in macrophage migration [19].

The role of M2 macrophages in endometriosis has remained controversial. IL-10, an anti-inflammatory cytokine associated with the M2 phenotype, is also elevated in women with endometriosis. IL-10 plays a critical role in eliminating unwanted cells and cellular debris. However, in a murine study, it was revealed that IL-10 administration promoted the growth of endometrial lesions. It was postulated that IL-10 suppresses immunity against endometrial implants that contribute to the development of endometriosis [47].

Taken together, we have shown that M2 macrophages emigrate more than M1 through the mesothelial monolayer, with higher C1P levels increasing the migration rate. Conversely, M1 macrophages are retained and secrete a variety of pro-inflammatory cytokines, shedding light on the role of ceramide-C1P metabolism in amplified and persistent peritoneal inflammation in endometriosis. These findings have far-reaching implications for our ability to screen, monitor, and treat endometriosis in patients. In terms of screening and monitoring, the levels of pro-inflammatory cytokines can be used as biomarkers of endometriosis and to predict its severity. Studies have shown the promise of using IL-1β and TNF-α for the prediction of endometriosis, with specificities of 0.85 and 0.72, respectively [48]. The upregulation of the CERK/C1P pathway and the inhibition of pro-inflammatory cytokine release are promising targets for the treatment of endometriosis (Figure 5). The activation of the C1P and CERK pathways to promote the migration and egress of M1 macrophages across the peritoneum may represent a novel therapeutic target to suppress the local pro-inflammatory environment that culminates in inflammation and the adhesions that are pathognomonic for peritoneal endometriosis disease. Finally, the M1 pro-inflammatory cytokines TNF-α and IL-1β were identified as promising targets in the treatment of endometriosis. In fact, a recent study revealed the potential use of fucoidan extracts from seaweed in inflammatory conditions. Fucoidan is a complex and sulfated biofunctional polymer rich in fucose that is found in brown seaweed and has been shown to reduce inflammation by negatively regulating pro-inflammatory cytokines such as TNF-α and IL-1β [49]. Recombinant human TNF binding protein 1, the activation of α-7 nicotinic acetylcholine receptors (nAChR), and the suppression of LPS-induced IL-1β mRNA expression have also been reported to be potentially effective endometriosis treatments [50,51].

## 4. Materials and Methods

### 4.1. Chemicals

All chemicals were purchased from Sigma (St. Louis, MO, USA) and cell culture reagents from Life Technologies (Carlsbad, CA, USA), unless otherwise stated.

### 4.2. Cell Culture

Human monocyte THP-1 cells (ATCC, Manassas, VA, USA) were cultured in RPMI medium (RPMI 1640; Gibco) supplemented with 10% heat-inactivated fetal bovine serum (HI FBS; Gibco); 1% penicillin-streptomycin (PS; Gibco); and 1% amphotericin B (ampB; Gibco), henceforth referred to as R10 medium. The FBS was heat-inactivated by being kept in a 56 °C water bath for 30 min. During passage, cells were pelleted at 300× *g* and quantified by staining with Trypan Blue (Sigma) and counting using a hemocytometer. Cells were seeded at 0.2 × 10^6^ cells/mL and grown to 80–100% confluence (0.8 × 10^6^ cells/mL to 1 × 10^6^ cells/mL). Cells were passaged every four to seven days. Cell viability was determined by staining with Trypan Blue and counting using a hemocytometer. THP-1 cells at passages 31–33 were used for experiments.

Transformed mesothelial cells (MeT-5A; ATCC) were grown in Medium 199 supplemented with 1.5 g/L sodium bicarbonate, 10% FBS, 1% PS, 3.3 nM epidermal growth factor (Corning, New York, United States), 400 nM hydrocortisone, 870 nM zinc-free bovine insulin, 20 mM HEPES, 3.87 µg/L selenious acid, and trace elements of B liquid (Corning, New York, United States), henceforth referred to as mesothelial medium. MeT-5A were seeded in T-25 or T-75 flasks (Nunc, ThermoFisher Scientific, Waltham, Massachusetts, United States) at a density of 6000 cells/cm^2^. Cells were grown to 70–80% confluence in a 37 °C humidified incubator at 5% CO_2_ and passaged every three to four days. Cells were trypsinized and pelleted at 300× *g*. Cell viability was assessed by staining with Trypan Blue and counting using a hemocytometer. Passage 13 and 14 cells were used for experiments.

### 4.3. Polarization of THP-1 Cells to M0, M1, and M2 Macrophages

THP-1 monocytes were differentiated to unactivated M0 macrophages and subsequently polarized to M1 and M2 as described by others, with some modifications [52,53]. Briefly, THP-1 monocytes were differentiated into M0 macrophages by phorbol 12-myristate 13-acetate (PMA) treatment. Next, 1.5 × 10^6^ THP-1 were seeded onto 6-well plates (Nunc) in 1.5 mL of R10 containing 25 nM PMA. After 48 h of incubation, the media were replaced with R0 (RPMI 1640 + 1% PS + 1% ampB) for 72 h. Subsequently, M0 cells were polarized to M1 and M2 with the respective cytokines in R2 (RPMI 1640 + 2% HI FBS + 1% PS + 1% ampB). M1 polarization: 20 ng/mL IFN-γ (R&D Systems) and 10 ng/mL LPS; M2 polarization: 40 ng/mL IL-4 (R&D Systems) and 40 ng/mL IL-13 (R&D Systems); M0 polarization: R2 alone was added. All polarization experiments were incubated for 48 h.

### 4.4. Transwell Migration of Macrophages

Sterile 6.5 mm transwell inserts with 5.0 μm or 8.0 μm pore polycarbonate membranes (Corning Transwell) were coated with 50 µL of 0.01% rat tail collagen type I (Corning) in 0.02 M acetic acid and incubated for 30 min at 37 °C. The coated membranes were washed twice with 200 µL 1 × PBS, and inserts were allowed to dry prior to use. Then, 0.5 × 10^5^ MeT-5A were seeded into each transwell insert in 200 µL of mesothelial medium (1.5 × 10^5^ cells/cm^2^). Care was taken to ensure that the lower chambers were left dry to prevent the premature migration of MeT-5A cells prior to the experiment. Cells were incubated for 24 h to form a confluent monolayer. A transwell insert was fixed after 24 h of seeding of MeT-5A with ice-cold methanol for 15 min. The inner membrane was stained with 100 µL crystal violet for 1 h. This served as a control to visualize and confirm that the mesothelial cells had grown to confluence and formed a monolayer.

Polarized macrophage cells were harvested by scraping and resuspended in migration media (50% M199 + 50% R2). Immediately prior to macrophage seeding, mesothelial media were removed from the MeT-5A monolayer in the transwell inserts. Then, 1 × 10^5^ macrophage cells in 200 µL migration media were added to each insert. M0, M1, M2 cells or 200 µL migration medium alone (blank) were added to the transwells, respectively. To the lower chamber, 700 µL of migration media was added. The transwells were incubated for 20 h to allow migration. After 20 h, the inner transwell membrane was washed in 1 × PBS, and the migrated cells were fixed in ice-cold methanol for 15 min. The cells were stained with Giemsa for 1 h then washed in milliQ water. Subsequently, the transwell membranes were cut out using a scalpel and mounted onto microscope slides using DPX mounting media and coverslips (VWR). Images at 20× magnification were acquired using an Olympus CKX41 microscope and Olympus cellSens imaging software (version 2.3). Migrated cells were counted using the Photoshop Portable counting tool. The total migrated cells per image were obtained and corrected for the basal migrated MeT-5A cells using the blank. The average number of M0, M1, or M2 macrophages migrated per image was obtained by averaging the replicates.

### 4.5. NVP-231-Inhibition of Macrophage Migration

The media of polarized M0, M1, and M2 macrophages were replaced with R0.5 media (RPMI 1640 + 0.5% HI FBS + 1% PS + 1% ampB). Cells were treated with 0, 200, 500, and 1000 nM of NVP-231 (Sigma-aldrich, Cat #, N9289), a potent and specific inhibitor of ceramide kinase (CERK) [32]. Immunoblot analysis was performed using anti-CERK antibodies (Abcam, Cat #, ab155061). After 2 h of incubation, macrophages were harvested and resuspended in migration media (50% M199 + 50% R2) with the respective molarity of NVP-231. Immediately prior to macrophage seeding, mesothelial media were removed from the MeT-5A monolayer in the transwell inserts. Then, 1 × 10^5^ macrophage cells in 200 µL migration media were added to the inserts (Figure 2A). M0, M1, M2 cells or 200 µL migration medium alone (blank) were added to the transwells, respectively. To the lower chamber, 700 µL of migration medium was added. The transwells were incubated for 20 h to allow migration. After 16 h, the inner transwell membrane was washed in 1 × PBS, and migrated cells were fixed in ice-cold methanol for 15 min. The cells were stained, and the transwell membranes were mounted and imaged as described in the previous section.

### 4.6. Multiplex Cytokine Analysis

After polarization, macrophage culture supernatants were collected and stored at −80 °C. Eight replicates each of the M0, M1, and M2 phenotypes were collected. The undiluted samples were randomized and assayed in duplicates for 45 cytokines, chemokines, and growth factors as previously described (Appendix A) [16,54]. A multiplex suspension bead immunoassay (45-Plex Human ProcartaPlex™ Panel 1, ThermoFisher. Cat #, EPX450-12171-901) was conducted according to the manufacturer’s instructions. Briefly, 10 μL of cell supernatant was mixed with 10 μL of antibody-conjugated, magnetic beads in a 96 DropArray plate (Curiox Biosystems) and rotated at 450 rpm on a plate shaker for 2 h in the dark at RT. The plate was washed three times with wash buffer (PBS + 0.05% Tween-20) on the LT210 Washing Station (Curiox) before adding 5 μL of secondary antibodies and rotating at 450 rpm for 30 min in the dark at RT. The plate was washed three times with wash buffer before 10 μL of streptavidin-phycoerythrin was added and rotated at 450 rpm for 30 min in the dark at RT. The plate was washed again three times with wash buffer, and 60 μL of reading buffer was subsequently added. The contents of the well were read using Bio-Plex Luminex 200 (BioRad). Beads were classified by the red classification laser (635 nm) into distinct sets. The green reporter laser (532 nm) excited the fluorescent reporter tag phycoerythrin. The quantitation of the 45 analytes in each sample was determined by extrapolation to a 6-point standard curve. Data analysis was carried out using five-parameter logistic regression modeling. Five factors (IL-13, IL-17A, MIP1α, MIP1β, and GM-CSF) with measurements missing from >50% of all samples were excluded. Cytokines that were added to the polarization media (IFN-γ, IL-4, and IL-13), even if they were bovine in origin, were excluded to prevent potential cross-talk. After excluding these factors, concentration levels of 38 cytokines remained (Appendix A). The assay coefficient of variation was 11.18%.

### 4.7. Liquid Chromatography Tandem Mass Spectrometry

Positive-ionization-mode liquid chromatography electrospray ionization tandem mass spectrometry (LC-ESI-MS/MS) in multiple reaction monitoring (MRM) mode using Triple Quadrupole 6460 with a Jet Stream (Agilent Technologies, Santa Clara, CA, USA) was applied for the relative quantification of sphingolipids as previously described, with some modifications [13,55]. Briefly, C_18_ reversed-phase LC (Zorbax Eclipse 2.1 × 50 mm I.D., 1.8 µm particle size; Agilent Technologies) was used to separate reconstituted lipids at 400 µL/min before entering the mass spectrometer. The dwell times were set at 10–50 ms per MRM transition. The MS parameters were as follows: gas temperature 250 °C, gas flow 10 L/min, sheath gas temperature 350 °C, sheath gas flow 11 L/min, capillary voltage 5000 V, and nozzle temperature 1500 V. The column thermostat and autosampler temperatures were maintained at 40 °C and 6 °C, respectively. The mobile phase consisted of 5 mM ammonium acetate in water (mobile phase A) and 5 mM ammonium acetate in methanol (mobile phase B). Sphingolipids were eluted using linear gradients from 60 to 100% B for 2 min, maintained at 100% B for 7 min, eluted by a linear gradient to 5% B for 2 min, held for another 4 min, and finally returned to 60% B for 2 min for the next injection. Deuterated internal standards corresponding to each sphingolipid class were verified for their absence and used for quantification against a nine-point calibration curve (1, 5, 10, 25, 50, 100, 250, 500, 1000 ng/mL). Data acquisition and processing were performed using MassHunter software (Agilent Technologies, USA, version 10). Sphingolipid quantification was performed using Agilent Quantitative software (version B.05). Only sphingolipids that had >100 intensity counts (approximately three times the limit of detection of our LC-MS instrument), and were present in >50% of the samples were retained for further analysis.

### 4.8. Statistical Analysis

For the secretome analysis, cytokines were analyzed by partial least squares regression (PLSR) using the nonlinear iterative partial least squares (NIPALS) algorithm (Unscrambler X version 10.1) after the normalization of data by performing log2 transformation. This was carried out to determine if there were specific cytokine signatures (cytokine being the x variable) that could explain the macrophage phenotype (outcome variable). Full cross-validation was applied in PLSR to increase the model performance by leaving out one sample at a time from the calibration dataset, and the model was calibrated on the remaining data points. Under cross-validation, a number of submodels were created based on all the samples that were not kept out in the cross-validation segment. For every submodel, a set of model parameters, including β-regression coefficients, was calculated. The β values of the regression coefficients associated with each x variable (cytokine) provided an indication of how strongly each x variable (cytokine) contributed to the macrophage phenotype (outcome variable). Additionally, for the concentration of each cytokine or sphingolipid, Student’s two-tailed, two-sample equal-variance *t*-test was used to make comparisons. The data analyzed were expressed as mean ± standard error, and *p*-values < 0.05 were considered significant.

## 5. Conclusions

Macrophages play a key role in the pathogenesis of peritoneal endometriosis. M1 macrophages, which secrete an increased pro-inflammatory cytokine milieu, demonstrated a lower innate migration rate across the mesothelium in vitro due to their lower C1P levels, plausibly leading to the development of chronic peritoneal inflammation in endometriosis. Thus, upregulating the CERK metabolic pathways and/or controlling macrophages to skew towards M2 phenotypes are potentially novel pharmacologic approaches for treating peritoneal endometriosis.

## Figures and Tables

**Figure 1 ijms-23-15977-f001:**
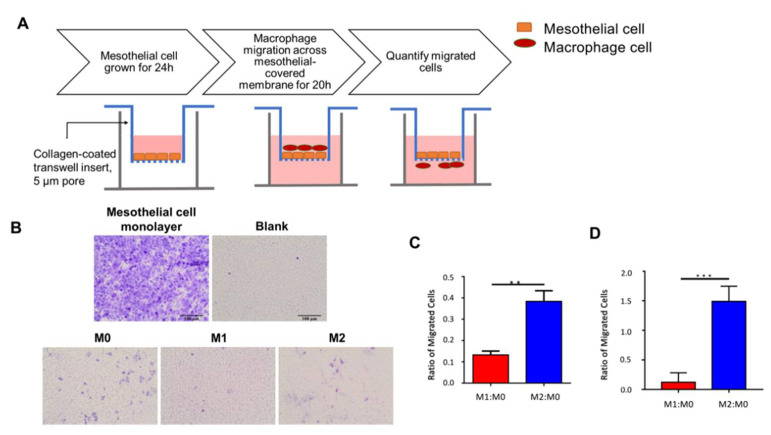
In vitro macrophage transwell migration. (**A**) Schematic of peritoneum macrophage in vitro migration model. MeT-5A mesothelial cells were seeded onto collagen-coated upper transwell membrane, incubated for 24 h, and allowed to form a mesothelial monolayer. Polarized macrophages were added to the upper transwell insert and allowed to migrate through the mesothelial cell layer and collagen-coated membrane for 20 h. Cells were fixed, and migrated cells on the outer transwell membrane were quantified. (**B**) Representative brightfield microscope images of migrated cells. MeT-5A mesothelial cells were grown on the upper membrane for 24 h to form a monolayer and crystal-violet-stained for visualization. Macrophages were allowed to migrate for 20 h; the migrated cells in the lower membrane were then Giemsa-stained. Scale bar = 100 µm. (**C**) Bar graph of mean M1:M0 and M2:M0 migration ratio obtained from three independent replicates. M1:M0 was significantly lower than M2:M0 (Student’s two-tailed *t*-test, *p* = 0.0012). (**D**) Bar graph of M1:M0 and M2:M0 migration with 30 nM C1P added to culture media. **, *p* < 0.01, ***, *p* < 0.005.

**Figure 2 ijms-23-15977-f002:**
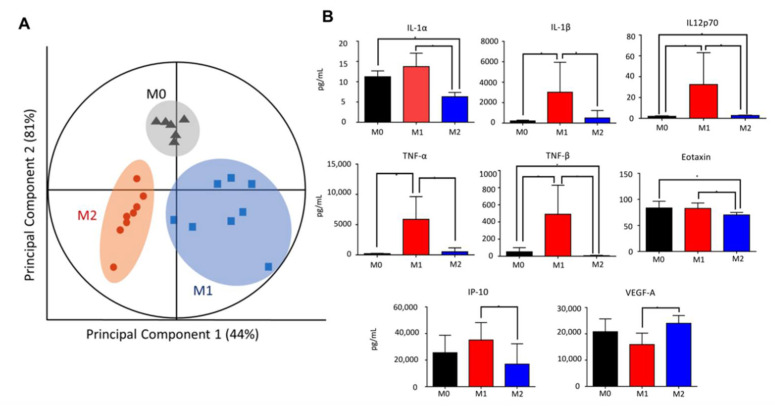
Distinct cytokine signatures of polarized macrophages. (**A**) Partial least squares regression (PLSR) score plot of culture supernatant suggesting distinct cytokine profiles of M0, M1, and M2. Each point displayed represents an independent sample, with eight samples each for M1 and M2 (N = 8). Two outliers were removed from sample M0 (N = 6). Clustered samples indicate a similar secretome and macrophage phenotype (circled). Each cluster is segregated from the other phenotypes, suggesting dissimilarity. (**B**) Cytokines significantly elevated in M1 compared to M2. Out of the 38 analytes assayed from the polarized cell supernatant, 13 analytes demonstrated significant differences in M1 versus M2 secretome (Student’s two-tailed *t*-test, * *p* < 0.05). Of these, IL-1α, IL-1β, TNF-α, TNF-β, and IL-12 are established M1 cytokines and are the strongest indicators of M1 polarization.

**Figure 3 ijms-23-15977-f003:**
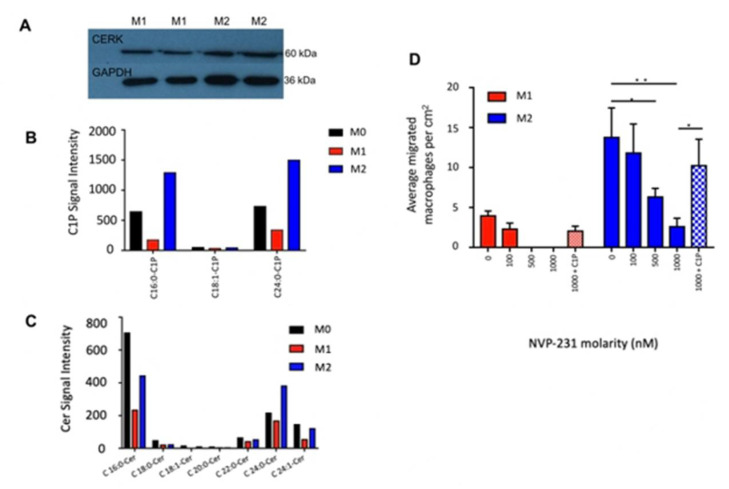
De novo synthesis of C1P in M2 macrophages induces emigration. (**A**) Immunoblot of CERK (60 kDa) in M1 and M2 macrophages. GAPDH (36 kDa) was used as loading control. (**B**) Intracellular ceramide-1-phosphate (C1P) levels were significantly higher in M2 than M1. (**C**) Intracellular ceramide (Cer) levels were significantly higher in M2 than M1. (**D**) CERK-inhibited macrophage migration. Bar graph of average number of migrated macrophage cells (per representative image taken) (N = 3). *, *p* < 0.05, **, *p* < 0.01.

**Figure 4 ijms-23-15977-f004:**
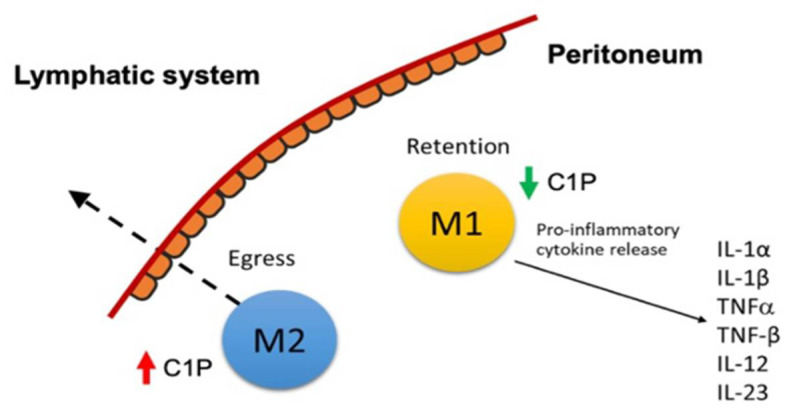
Schematic of M1 macrophage retention in the peritoneum.

**Figure 5 ijms-23-15977-f005:**
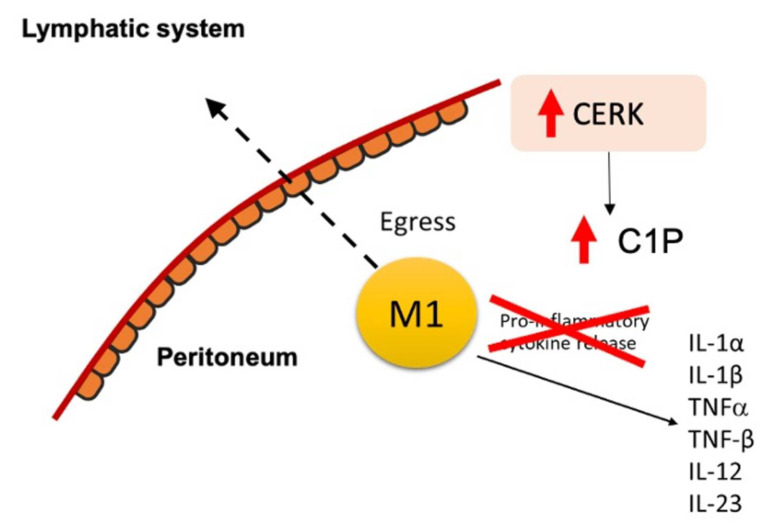
Schematic of potential novel pharmacologic treatment for endometriosis that targets the CERK/C1P pathways and inhibits pro-inflammatory cytokine release from M1 macrophages.

## Data Availability

The data presented in this study are available on request from the corresponding author. The data are not publicly available due to privacy issues.

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
