# Peer review of "Decreased Innate Migration of Pro-Inflammatory M1 Macrophages through the Mesothelial Membrane Is Affected by Ceramide Kinase and Ceramide 1-P"

_ijms, 2022, doi:10.3390/ijms232415977_

Round 1
Reviewer 1 Report
This study by Ku and coworkers aimed to ascertain the differences in migratory properties of macrophage phenotypes across mesothelial cells in vitro, and the correlation with ceramide-1-phosphate (C1P) levels. An in-vitro model of macrophage migration across a peritoneal mesothelial cell monolayer membrane was developed. The results would indicate that M1 macrophages are more sessile, emigrating 2.9-folds lower than M2 macrophages. The M1 macrophages displayed a pro-inflammatory cytokine signature, including IL-1α, IL-1β, TNF-α, TNF-β and IL-12p70. Decreased levels of C1P, an inducer of migration were found in M1 macrophages. C1P is generated by ceramide kinase (CERK) from ceramide, and blocking C1P synthesis via the action of NVP231, a specific CERK chemical inhibitor, abolished the emigration of M1 and M2 macrophages by up to 6.7-folds.
The study is interesting and the manuscript is clear and well-written. However, the only major issue is that the study is completely focused on the macrophages-mesothelial interaction and not on endometriosis. The model can be applied to every condition in which there is a peritoneal inflammation. The model does not include cells from endometriosis, endometriotic implants or any biological samples from endometriosis women. The study uses cell lines for the experiments of mesothelial cells cocultured with macrophages. Moreover, macrophages are polarized according to previously published protocols.
Therefore, the structure of the paper should be rewritten completely (Abstract, Introduction and Discussion) in order to focus on this biological model and then consider to discuss it in the context of endometriosis. Endometriosis can be only a supposed condition where this model might function and not the focus of the paper.
Moreover, the rationale of the study to support phagocytosis of endometriotic cells by macrophages is not so strong. The reference reported by the authors is dated 1991.Certainly, macrophages are involved in endometriosis but the exact mechanisms exerted are not well understood and may be different. This further supports the need for a change in the paper structure.
Reviewer 2 Report
In the manuscript ‘Decreased innate migration of pro-inflammatory M1 macrophages through the mesothelial membrane is affected by ceramide kinase and ceramide 1-P’the authors evaluated the differences in migratory properties of M1 and M2 macrophages across mesothelial cells in vitro models including 2D and 3D cultures of endometrial stromal and peritoneal mesothelial cells. The study has a relevant theme, however, the paper needs major revision before accepting for publication. The authors did not avoid some, which are listed below in the points:
Line33-35 –Where the implants are present, please specify
Line 38-40-What the authors mean by micro and macroenvironment? Please explain.
Line 45 – Please, add references.
Line 82 – The sentence ‘In an in vitro’ will be a better sound without the word ‘In an’. Finally, I propose the sentence starting like this: ‘In vitro study….’.
Line 102 – Please, correct punctuation errors, and remove the dot.
Line 115, 120, 121, etc. – The P-Value should be written in italics. Please, correct this mistake in all manuscripts.
Line 123 – Please, correct the text in the figures, it is illegible. ( Figure 1. C, 1.D; Figure 2. B; Figure.3B,3C,3D).
Line 124, 125, etc. – Please, correct the ‘in vitro’ expression in the manuscript, it should be written in italics.
Line 112, 113, 126, 128, 130, etc. – Please standardize the way of writing units in the manuscript, it is about the hours, degrees Celsius. One time is e.g. 20 h, but another time is e.g. 20h.
Line 134 – Please, correct the punctuation errors in the manuscript.
Line 141 – Please explain the abbreviations used the first time in the manuscript.
Line 147 – Correct the punctuation errors in the manuscript.
Line 160 – Please, correct the text in the figures, it is illegible.
Line 166 – Please, replace the expression ’13 of 38 analytes….’ For ‘The 13 of 38 analytes…’
Line 197 – Please, correct the text in the figures, it is illegible.
Line 198 – Please, correct the ‘de novo’ expression in the manuscript, it should be written in italics.
Line 205 – Please, correct the “4” should not be written in italics.
Line 210, 216, 247, 277, etc. – Please, correct the editorial errors in the manuscript.
Line 288 – ‘All chemicals were purchased from Sigma….’ – What exact reagents were used in the research?
Line 292, 293, etc – Please, add the country of the company in the manuscript, e.g. Sigma, USA.
Line 312 – Please, add the space before the bracket.
Line 341 – Please correct the sentence ‘The cells were stained with Giemsa or 1 h..’ is correct. Shouldn’t ‘or’ be replaced with ‘for’?
Line 366 – Please, unify the place of figures in front of references in the test.
Line 391 – Please, add the space before the bracket.
Line 549 – It is not the number doi on this reference. Please add the number doi.
Line 583, 586 – Please, correct the names of the journals, either the full names or the abbreviations.
Line 612 – This line is unnecessary.
Round 2
Reviewer 2 Report
Thank you the Authors for improving the paper.
Now I recommend the manuscript for publication.
Author Response
Thank you for taking your time to review our work and for recommending it for publication.